# Muons in showers with energy $E_0 \geq 5$ EeV and QGSjetII-04 and EPOS LHC models of hadronic interactions. Is there a muon deficit in the models?

Stanislav Knurenko and Igor Petrov⋆

Yu. G. Shafer Institute of Cosmophysical Research and Aeronomy, Yakutsk, Russia

⋆ igor.petrov@mail.ysn.ru

## Abstract

The paper presents data on the muon component with a threshold $\varepsilon_{thr} \geq 1$ GeV. Air showers were registered at the Yakutsk array during almost 50 years of continuous air shower observations. The characteristics of muons are compared with calculations of QGSjetII-04 and EPOS LHC models for a proton and an iron nucleus. There is a muon deficit in the models, at energies greater than 5 EeV. To make an agreement between experimental data and simulations on muons, further tuning of the models is required.

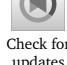

## 1 Introduction

The study of cosmic rays (CR) with highest energies greater than 5 EeV is very important from the nature of source perspective, propagation and interaction with matter and magnetic fields in outer space [1,2]. The properties of CR in these energy ranges are not well known and, for this reason, they are the subject of research at large air shower experiments.

There are two aspects of CR properties: astrophysical and nuclear physics. The astrophysical aspect includes the study of the spectrum, mass composition and anisotropy of CR. The recently obtained irregularity in the CR spectrum at an energy of $\sim 10^{17}$ eV [3–5], which is associated with the rigidity of particles of galactic origin, is of even greater interest, because the possibility of determining the boundary of the transition from galactic to extragalactic cosmic rays. The knowledge of the mass composition of primary particles in the region of $10^{16} - 10^{19}$ eV can help to establish the boundary between galactic and extragalactic cosmic rays [6–9].

The nuclear physics aspect includes the study of the interaction of primary particles with the nuclei of air atoms. First — obtaining the characteristics of an elementary event: the cross

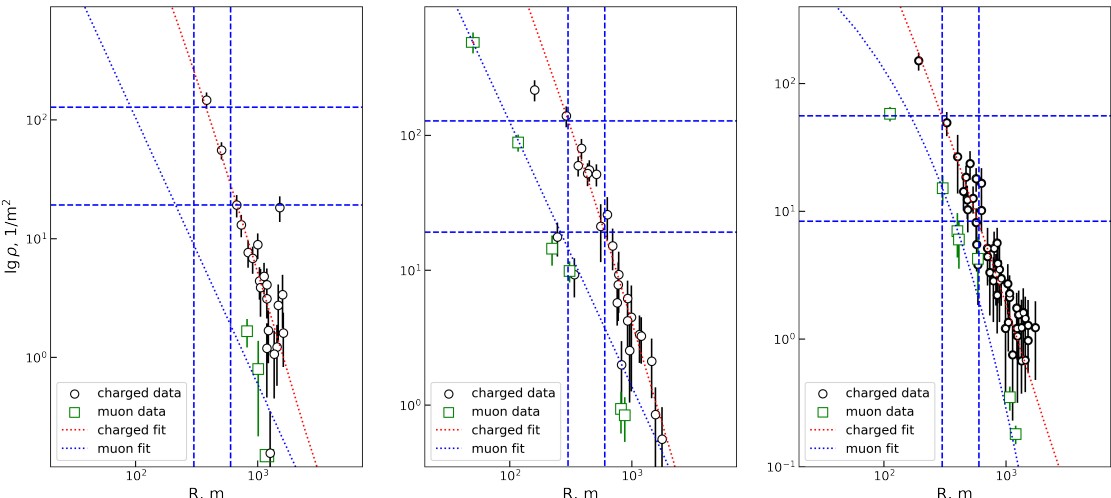

Figure 1: Individual showers with muons. Left: Date = 2018-01-04, $\lg E_0 = 19.23$ eV, $\theta = 25.8°$, $\phi = 213.6°$, $\lg \rho_s = 1.45$ m$^{-2}$. Center: Date = 2018-01-05, $\lg E_0 = 19.32$ eV, $\theta = 44.8°$, $\phi = 303.5°$, $\lg \rho_s = 1.28$ m$^{-2}$. Right: Date = 2014-01-22, $\lg E_0 = 19.03$ eV, $\theta = 46.3°$, $\phi = 108.5°$, $\lg \rho_s = 0.92$ m$^{-2}$.

section of inelastic interaction $\sigma_{A-air}$, the average coefficient of inelasticity $\langle K_{in} \rangle$ and the multiplicity of secondary particles $n_{ch}$ - hadrons according to air shower registration data in the ultrahigh energy region [10–13]. Second — the study of the processes of interaction and decay of high-energy hadrons and the influence of these processes on muon and electromagnetic cascades in the development of air showers. To determine most of the listed characteristics, including the estimate of the atomic weight of the primary particle, it is required to know the theoretical model of hadronic interactions that describes the development of the nuclear component of a real air shower. The available models, as shown by comparison with experimental data on muons, cannot yet quantitatively describe the excess of muons in measured air showers. Therefore, the aim of this work is to test modern models of hadronic interactions using data on muons in individual showers.

## 2 Lateral distribution of charged particles and muons in showers with energy greater than 10 EeV

At the Yakutsk array, the fluxes of charged particles and muons are measured by scintillation detectors with an area of 2 m$^2$. The energy threshold of the detectors are 1.8 and 10 MeV. Observation stations within the detector array are located in such a way that showers always contain information about muons within distances of 100–1300 m from the shower axis. After mathematical processing of the data, the air shower lateral distribution functions (LDF) are plotted from the calibrated signals. As an example, Fig. 1 shows the LDF of three individual showers with energies greater than 10 EeV and different zenith angles $\theta$. The curved lines in the figures are approximations of each of the air shower components [14, 15].

### 2.1 Correlation of $\rho_\mu(600)$ with air shower energy

Fig. 2 shows the averaged $\langle \rho_\mu(600) \rangle$ values as a function of energy for different zenith angles: $\cos \theta = (0.667\text{-}0.834)$ and $\cos \theta = (0.834\text{-}1.000)$. Where, $\rho_\mu(600)$ is the muon flux density obtained at the 600 m distance from the shower axis. In addition, calculations by the QGSjetII-

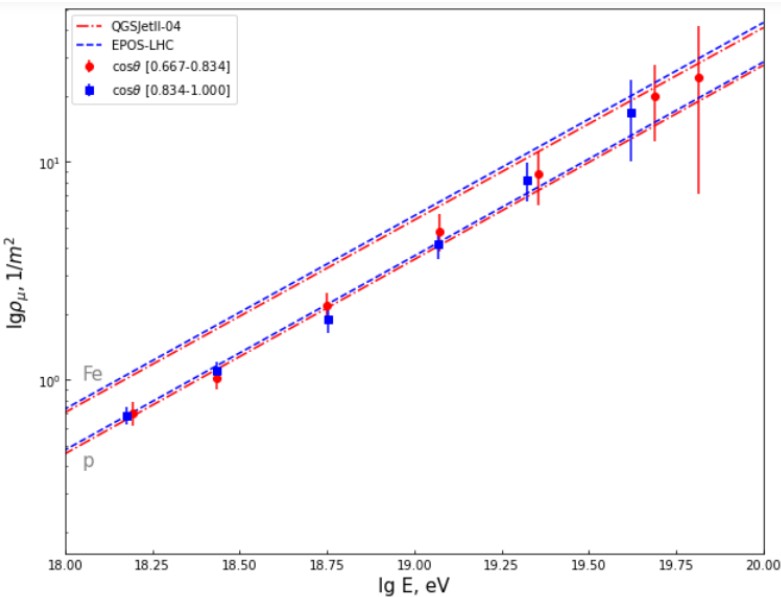

Figure 2: Dependence of muon flux density $\langle \rho_\mu(600) \rangle$ on energy $E_0$. Lines are calculation for QGSjetII-04 and EPOS LHC for proton and iron nucleus.

04 and EPOS LHC models for the proton and iron nucleus are plotted there. The shower samples are consistent with each other within the statistical errors, which indicates the absence of systematic errors due to the inclusion of showers with different zenith angles in the total sample. Comparison of the experimental data with calculations shows that in the range 1-10 EeV the data points lay on the calculations for the proton, and starting from an energy of 20 EeV, the data points get closer to the calculations for the iron nucleus. Although the errors in this energy range are large and do not exclude agreement with the calculations for the proton. The agreement between the models and experimental data on $\rho_\mu(600)$, taking into account the mass composition of cosmic rays with respect to the parameter z [16], is given below. For a correct comparison of the results of the analysis of the parameter z obtained at different experiments, it is necessary to agree on the energy estimation between the experiments. In our opinion, this can be done using formula (1), where $E_0$ is determined by the parameter $\rho_\mu(600)$ [17]

$$\lg E_0 = 18.33 + 1.12 \cdot \lg(\rho_\mu(R = 600)). \tag{1}$$

## 2.2 $\rho_\mu(600)/\rho_{\mu+e}(600)$ correlation with depth of maximum of electron-photon component of air shower

Longitudinal development of ultrahigh-energy air showers is reconstructed at the Yakutsk array from measurements of the Cherenkov light LDF [10, 18] and directly using track Cherenkov detectors [19, 20]. These methods make it possible to determine the characteristics of the air shower development cascade curve $X_{max}$.

We selected individual air showers with measured Cherenkov light, charged and muon component data. We reconstructed $X_{max}$ — depth of maximum by Cherenkov light data and estimated $\rho_\mu(600)/\rho_{\mu+e}(600)$ by charged and muon components data. Then the data were binned by fraction of muons for three different zenith angles and for each bin we determined $\langle X_{max} \rangle$. Fig. 3 shows a correlation of the parameter $\rho_\mu(600)/\rho_{\mu+e}(600)$ with $X_{max}$ for three zenith angles $< \theta >= 18°$. $< \theta >= 32°$ and $< \theta >= 58°$.

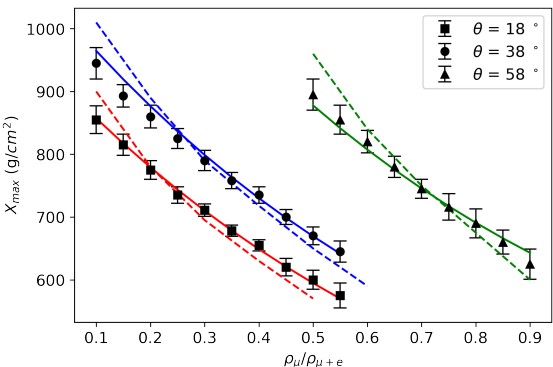

Figure 3: Relationship between the depth of the electromagnetic cascade maximum and the fraction of muons at a distance of 600 m from the shower axis. Calculation curves according to the model of hadronic interactions QGSjetII-04 for zenith angles $< \theta >= 18° < \theta >= 32°$ and $< \theta >= 58°$.

Using obtained data (Fig. 3) and exponential function, an empirical relationship between $\rho_\mu(600)/\rho_{\mu+e}(600)$ and $X_{max}$ was found:

$$X_{max} = (745 + 413 \cdot (\sec\theta - 1)) \cdot \exp\left(-\frac{\rho_\mu(600)/\rho_{\mu+e}(600)}{-0.818 - 0.037 \cdot (\sec\theta - 1)}\right)$$
$$+ (172 + 132 \cdot (\sec\theta - 1)). \tag{2}$$

Coefficients in the equation (2) were determined by approximation of the experimental data.

Further, formula (2) was used to calculate $X_{max}$ in individual showers using the parameter $\rho_\mu(600)/\rho_{\mu+e}(600)$. This method for estimating $X_{max}$ increased the statistics of showers with determined $X_{max}$ for this work. For example, the measurement time for muons at the Yakutsk array is (50-60)%, and the measurement time for Cherenkov light is ∼(6-10)% of the total time of charged particles measurements. In addition, this technique does not depend on weather conditions, while the registration of Cherenkov light depends on the transparency of the atmosphere, the presence of the Moon in the sky, the Northern Lights, and other factors.

## 2.3 Dependence of $X_{max}$ on air shower energy. Analysis of $\rho_\mu(600) - E_0$ and $X_{max} - E_0$ correlation in the frame of QGSjetII-04, EPOS LHC

Fig. 4 compares the average $X_{max}$ values obtained from the muon component with the $X_{max}$ results obtained from the other components at the Yakutsk [9] and other experiments: PAO [21], TA [4], LOFAR [22] and Tunka [23].

The muon component of the air shower is considered to be the most sensitive to hadronic interactions. For this reason, muons are usually used to test different models in order to select a model that describes the development of real air showers. Comparison of the number of muons detected in experiments with earlier calculations [24–26] indicated a deficit of muons at the energies greater than $10^{17}$ in almost all models. The discrepancy increases with energy up to ∼30%. Improvement of the models [27–29] led to a convergence of the experiment with the new models, but the difference could not be completely eliminated [30]. In the present work, we compare the muon component detected at the Yakutsk array with modern QGSjetII-

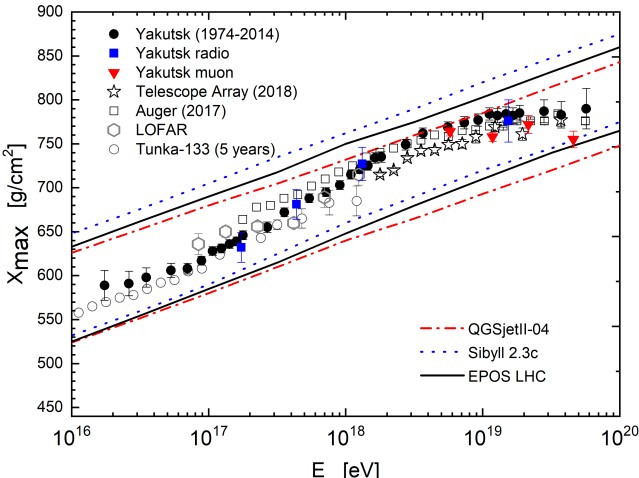

Figure 4: Dependence of air shower depth of maximum on energy. Results of small and large experiments are presented. Lines are simulations for different hadron interaction models for primary proton and iron nucleus.

04 and EPOS LHC models for energies greater than 5 EeV. The z parameter was used to test the models:

$$z = \frac{\ln \rho_\mu^{exp} - \ln \rho_\mu^{p}}{\ln \rho_\mu^{Fe} - \ln \rho_\mu^{p}}, \tag{3}$$

where $\rho_\mu^{exp}$ — muon flux density at 600 m from air shower axis, p, Fe — calculated $\rho_\mu$ by QGSjetII-04 and EPOS-LHC, for proton and iron nucleus [16]. The z-value is computed relative to hadronic interaction model.

In formula (3), the parameter z is expected to be in range between 0 and 1, where 0 is pure proton showers and 1 is pure iron showers.

The obtained value of z is shown in Fig. 5 for two models QGSjetII-04 [27] and EPOS LHC [28]. Contour lines show the boundaries of errors in determining the parameter z. In addition there is an expected $z_{mass}$ value shown in gray contour, estimated from $X_{max}$ of Cherenkov light measurements [9]. $z_{mass} = \frac{\langle \ln A \rangle}{\ln 56}$ according to [16].

To assess the accuracy of the result, we used the error functional for measuring the muon flux density (4), which was established empirically in the course of analyzing the operation of adjacent scintillation detectors, followed by verification of the result obtained by the Monte Carlo method [31,32]:

$$D_\mu = \sigma^2(\rho_\mu^{exp}) = (\rho_\mu^{exp})^2 \cdot \left( \beta^2 + \frac{1 + \alpha^2}{s \cdot \rho_\mu^{exp} \cos \theta} \right), \tag{4}$$

where $\beta^2$ — relative error of amplitude meausrements, which reflects instrumental fluctuation of scintillation detector response, $\alpha^2$ — statistical error, according to Poisson distribution.

The results of testing the QGSjetII-04 and EPOS-LHC models using muon data are shown in Fig. 5. As can be seen from Fig. 5, the results agree within the experimental errors with the predictions from the QGSjetII-04 and EPOS LHC models up to energies of 10 EeV. Above 10 EeV, there is a trend for an increase of z-value. The total error (systematic and statistical) of the z parameter for the data from the Yakutsk array was determined by differentiating

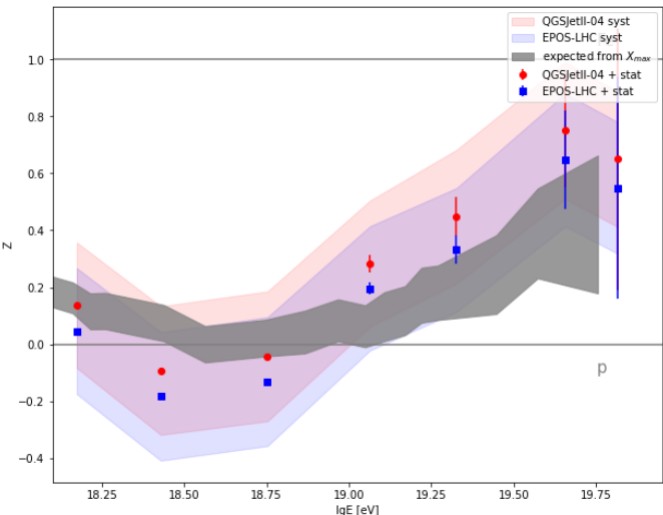

Figure 5: Testing of the QGSjetII-04 (blue squares) and EPOS LHC (red dots) models by the parameter $\rho_\mu(600)$. Gray contour is expected $z_{mass}$ value from $X_{max}$ [9].

expression (3) with respect to three parameters. As a result, it turned out to be equal to $\sim$17% and is shown in Fig. 5 by a filled contour for two models.

The origin of such a behaviour of the z-value that has emerged from experimental data is not clearly known. It could be because of slowing down the development of air shower in the atmosphere for energies greater than 5 EeV, as $X_{max}$ value shifts to sea level more slowly than in the energy range $(0.1-5)$ EeV [9]. Or with a difference in absorption ranges of muons with threshold $\varepsilon_{thr} \geq 1$ GeV$\Lambda\mu$ in models and found from experimental data [18]. The latter is possibly related to the problem in describing the energy spectra of muons at the beginning of the nuclear cascade of air shower development [33, 34]. It is possible that a sharp change in the mass composition can also lead to such a behavior of the z function. As can be seen from Figs. 4, the experimental data indicates a change in the mass composition from light at lower energies to a heavier composition starting from an energy of 5 EeV. In any case, to solve this problem, further research is required at experiments studying air showers of highest energies, including additional experiments at the LHC, such as oxygen beam collisions [35]

## 3   Conclusion

The Yakutsk array has been a testing ground for the study of cosmic radiation in the field of ultra-high and highest energies for 50 years. The complex registration of charged particles, muons, Cherenkov light and air shower radio emission, together with the developed software for the preliminary and subsequent analysis of showers, made it possible to study the radial and longitudinal development of showers and determine their main characteristics [12, 18, 32, 36, 37]. Large statistics of showers with a good precision made it possible to create a database in the energy range $10^{15} - 10^{20}$ eV for the study of air shower physics.

Long-term registration of the muon component at the Yakutsk array has shown that the muon flux density at a distance of 600 m from the shower axis $\rho_\mu(600)$ is proportional to the shower energy $E_0$, and the ratio $\rho_\mu(600)/\rho_{\mu+e}(600)$ is related to the longitudinal development of air showers — $X_{max}$. An analysis of individual showers based on the muon component in the energy range above 5 EeV showed that, within the framework of the QGSjetII-04 and EPOS LHC models, the composition of cosmic rays begins to slowly change towards medium nuclei

and, above at energies of 20 EeV, becomes heavier with respect to the energy range 0.1-2 EeV.

From comparison of z-value with $z_{mass}$ estimated by $X_{max}$ measurements (Fig. 5) we can assume that up to energies 10 EeV there is no muon deficit. Although, for energies greater than 10 EeV there is a trend for z-value increase, but since it's within pure iron composition and because of high systematic uncertainties we can't confirm a muon deficit. If we assume that there is a deficit of muons in the models [38], then this fact requires a further analysis and explanation. For example direct comparison of individual air showers with simulations.

## Acknowledgements

**Funding information**   This work was carried out in the framework of research project No. AAAA-A21-121011990011-8 by the Ministry of Science and Higher Education of the Russian Federation.

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
