# Peer review of "Muons in showers with energy E0 5 EeV and QGSjetII-04 and EPOS LHC models of hadronic interactions. Is there a muon deficit in the models?"

_SciPost Physics Proceedings, doi:SciPost Phys. Proc. 13, 032 (2023)_

## Round 1 · Referee Report · Anonymous (Referee 1) · 2022-8-30

Strengths
1 - This paper reports the measurement of the muon content in air showers with energies above 5 EeV and compares the results with predictions from the current hadronic interaction models EPOS-LHC and QGSJet-II.04.
2 - Given the discrepancies observed in muon measurements by most recent experiments, this work represents an important contribution to understanding the origin of the Muon Puzzle in air showers.
2 - Given the discrepancies observed in muon measurements by most recent experiments, this work represents an important contribution to understanding the origin of the Muon Puzzle in air showers.
Weaknesses
1 - The interpretation of the results in terms of the z-values has significant flaws which should be addressed before publication in SciPost proceedings.
Report
This paper reports the measurement of the muon content in air showers with energies above 5 EeV and compares the results with predictions from the current hadronic interaction models EPOS-LHC and QGSJet-II.04. Given the discrepancies observed in muon measurements by most recent experiments, this work represents an important contribution to understanding the origin of the Muon Puzzle in air showers. However, the interpretation of the results has some significant flaws which should be addressed by the authors before publication in SciPost proceedings.
In particular, in Section 2.3 (and in the conclusions) the measurements are discussed in terms of the z-values, i.e. Eq. (3). The current text reads "In formula (3), the parameter z reflects the difference between the experimental data and the expected value of muons for one or another model of hadronic interactions. A positive value of z indicates an excess of muons in the experimental data compared to the considered model. A zero value indicates the complete agreement between the experiment and the calculations.". This statement is *not* true. Instead, the z-value is in fact constructed such that an observation of a muon density of z=0 is consistent with simulated proton showers and z=1 is consistent with simulated iron showers with a given hadronic interaction model (as obvious from Eq. (3), see also H. Dembinski et al., EPJ Web. Conf. 210 (2019)). Thus, a positive value of z does *not* indicate an excess of muons with respect to model predictions. Hence, all subsequent conclusions are flawed. In fact, the measurements presented in this work are bracketed by the proton and iron predictions within their uncertainties. Since this is generally a physically allowed region for air shower measurements, the conclusion that discrepancies between experimental data and model predictions are observed above 5 EeV does not hold. In order to draw conclusions from the given distributions, the z-values should instead be compared to current composition measurements, such as measurements of the maximum shower depth, Xmax, or predictions from recent cosmic ray flux models, for example. This was done recently in Soldin et al., PoS(ICRC2021)349 (2021), for example. However, given the lack of such a comparison, the current manuscript does not allow to draw the current conclusions. In fact, the description of the current conclusions is not valid due to the different meaning of the z-values. These important issues should be addressed before publication in SciPost.
In addition, I have some minor comments and suggestions, mainly on wording:
Page 1/2: "Second — the study of the processes of interaction and decay of high-energy hadrons and the influence of these processes on muon and electromagnetic cascades in the development of air shower" -> "... in the development of an air shower." or "... in the development of air showers.".
Page 2: "The energy threshold of the detectors are 10 and 1.8 MeV". I suggest to write "The energy threshold of the detectors are 1.8 MeV and 10 MeV.".
Page 2 (and throughout the paper): For the average muon densities, <rho_mu(600)>, I highly recommend to use \langle and \rangle instead of "<" and ">". Also, I suggest to write explicitly that <rho_mu(600)> is the muon density obtained at 600m from the air shower axis.
Page 3: "... taking into account the mass composition of cosmic rays with respect to the parameter z, ...". Please add a reference to the z-paramater, e.g. Dembinski et al., EPJ Web. Conf. 210 (2019) or F. Gesualdi et al., PoS(ICRC2021)473 (2021) as the z-values are only introduced later in the text.
page 3: "We used a sample of showers with all three components: fluxes of charged particles, muons, and Cherenkov light, were measured simultaneously." I do not understand what is meant by "all three components", i.e. an air shower includes more than these three components. Moreover, muons are also charged particles. I recommend to clarify the description in the text.
Page 4: Is there a reference for Eq. (2)? Otherwise, I believe it needs more clarification on how this formula was obtained.
Page 4: "Comparison of the number of muons detected in experiments with earlier calculations [23–25] indicated a significant deficit of muons up to ∼30% in almost all models." I highly suggest to mention the cosmic ray energy range here since the muon excess seems to be an energy-dependent effect, see e.g. Dembinski et al., EPJ Web. Conf. 210 (2019) or Soldin et al., PoS(ICRC2021)349 (2021).
Page 6: "...including the continuation of the muon measurement
experiment at the LHC." Which muon measurement experiment at LHC? Please add a reference.
References: There are several references where the format of the "et al." in the author list is nor correct, I believe. E.g. "M. Dyakonov, A. Ivanov, S. Knurenko and et al." should be "M. Dyakonov, A. Ivanov, S. Knurenko et al." I believe (no "and" before et al.). This also holds for other references.
In particular, in Section 2.3 (and in the conclusions) the measurements are discussed in terms of the z-values, i.e. Eq. (3). The current text reads "In formula (3), the parameter z reflects the difference between the experimental data and the expected value of muons for one or another model of hadronic interactions. A positive value of z indicates an excess of muons in the experimental data compared to the considered model. A zero value indicates the complete agreement between the experiment and the calculations.". This statement is *not* true. Instead, the z-value is in fact constructed such that an observation of a muon density of z=0 is consistent with simulated proton showers and z=1 is consistent with simulated iron showers with a given hadronic interaction model (as obvious from Eq. (3), see also H. Dembinski et al., EPJ Web. Conf. 210 (2019)). Thus, a positive value of z does *not* indicate an excess of muons with respect to model predictions. Hence, all subsequent conclusions are flawed. In fact, the measurements presented in this work are bracketed by the proton and iron predictions within their uncertainties. Since this is generally a physically allowed region for air shower measurements, the conclusion that discrepancies between experimental data and model predictions are observed above 5 EeV does not hold. In order to draw conclusions from the given distributions, the z-values should instead be compared to current composition measurements, such as measurements of the maximum shower depth, Xmax, or predictions from recent cosmic ray flux models, for example. This was done recently in Soldin et al., PoS(ICRC2021)349 (2021), for example. However, given the lack of such a comparison, the current manuscript does not allow to draw the current conclusions. In fact, the description of the current conclusions is not valid due to the different meaning of the z-values. These important issues should be addressed before publication in SciPost.
In addition, I have some minor comments and suggestions, mainly on wording:
Page 1/2: "Second — the study of the processes of interaction and decay of high-energy hadrons and the influence of these processes on muon and electromagnetic cascades in the development of air shower" -> "... in the development of an air shower." or "... in the development of air showers.".
Page 2: "The energy threshold of the detectors are 10 and 1.8 MeV". I suggest to write "The energy threshold of the detectors are 1.8 MeV and 10 MeV.".
Page 2 (and throughout the paper): For the average muon densities, <rho_mu(600)>, I highly recommend to use \langle and \rangle instead of "<" and ">". Also, I suggest to write explicitly that <rho_mu(600)> is the muon density obtained at 600m from the air shower axis.
Page 3: "... taking into account the mass composition of cosmic rays with respect to the parameter z, ...". Please add a reference to the z-paramater, e.g. Dembinski et al., EPJ Web. Conf. 210 (2019) or F. Gesualdi et al., PoS(ICRC2021)473 (2021) as the z-values are only introduced later in the text.
page 3: "We used a sample of showers with all three components: fluxes of charged particles, muons, and Cherenkov light, were measured simultaneously." I do not understand what is meant by "all three components", i.e. an air shower includes more than these three components. Moreover, muons are also charged particles. I recommend to clarify the description in the text.
Page 4: Is there a reference for Eq. (2)? Otherwise, I believe it needs more clarification on how this formula was obtained.
Page 4: "Comparison of the number of muons detected in experiments with earlier calculations [23–25] indicated a significant deficit of muons up to ∼30% in almost all models." I highly suggest to mention the cosmic ray energy range here since the muon excess seems to be an energy-dependent effect, see e.g. Dembinski et al., EPJ Web. Conf. 210 (2019) or Soldin et al., PoS(ICRC2021)349 (2021).
Page 6: "...including the continuation of the muon measurement
experiment at the LHC." Which muon measurement experiment at LHC? Please add a reference.
References: There are several references where the format of the "et al." in the author list is nor correct, I believe. E.g. "M. Dyakonov, A. Ivanov, S. Knurenko and et al." should be "M. Dyakonov, A. Ivanov, S. Knurenko et al." I believe (no "and" before et al.). This also holds for other references.

---

## Round 2 · Referee Report · Anonymous (Referee 1) · 2022-9-13

Report

I would like to thank the authors very much for considering my comments and suggestions. The manuscript clearly improved in clarity and content. However, I have some remaining remarks that should be considered before publication in SciPost proceedings.

While I highly appreciate the change of wording in the description of the discrepancies between models and data, in particular in Section 2.3 and the conclusions, it seems to me that the abstract needs some minor modifications as well since it still reports "There is a muon deficit in the models...". In fact, this is in contradiction with the new wording in the conclusions, "...we can't confirm a muon deficit.".

In this context, it may be worthwhile not only to discuss the trend of the z-values to be too high, i.e. a deficit in simulations, but also add some comments on the behavior below 10EeV where an opposite trend can be observed (which seems to be of similar size).

In addition, I only have a few very minor suggestions. Please feel free to ignore them if you disagree:

General: The figure labels are generally very small. The figures would improve from a larger font size.
p1: There is an ugly line break in "10^16-10^19 eV", i.e. the unit appears in the next line. Maybe this can be fixed.
p2: "Observation stations on array plane..." I believe this needs some rewording, e.g. "Observation stations within the detector array..." or similar.
p5: "Such behaviour of z-value..." I believe this sentence needs some re-wording, for example "The origin of such a behaviour of the z-value...".
p6: "Large statistics of showers with a good "history" made it possible..." I do not understand what "good history" means. Also, it does not sound very scientific. Thus, I recommend a re-wording or clarification here.
p6: "...up to energies (of) 10EeV there is no muon deficit." -> "...muon deficit in the models."

---

## Round 2 · List of Changes

Replaced Fig. 5 with updated one.

Updated Fig. 5. description.

Changed z-value description in section 2.3.

Corrected conclusion according to referee suggestions.

Added 2 additional bibliography, namely H. Dembinski et al. EPJ Web of Conf. 210 (2019) and J. Albrecht et al. Astrophys. Space. Sci. 367 (2022)

Corrected wording in some sentences

Corrected references

Minor corrections throughout the text

---

## Round 3 · List of Changes

p1: There is an ugly line break in "10^16-10^19 eV", i.e. the unit appears in the next line. Maybe this can be fixed.
Fixed line break

p2: "Observation stations on array plane..." I believe this needs some rewording, e.g. "Observation stations within the detector array..." or similar.
Changed the wording

p5: "Such behaviour of z-value..." I believe this sentence needs some re-wording, for example "The origin of such a behaviour of the z-value...".
Changed the wording

p6: "Large statistics of showers with a good "history" made it possible..." I do not understand what "good history" means. Also, it does not sound very scientific. Thus, I recommend a re-wording or clarification here.
Changed to "good precision"

---

## Editorial Decision

published